# Remodeling Osteoarthritic Articular Cartilage under Hypoxic Conditions

**DOI:** 10.3390/ijms23105356

**Published:** 2022-05-11

**Authors:** Mohd Heikal Mohd Yunus, Yemin Lee, Abid Nordin, Kien Hui Chua, Ruszymah Bt Hj Idrus

**Affiliations:** 1Department of Physiology, Universiti Kebangsaan Malaysia, Cheras, Kuala Lumpur 56000, Malaysia; ckienhui@gmail.com (K.H.C.); ruszyidrus@gmail.com (R.B.H.I.); 2MedCentral Consulting, Jalan 27/117A, Bandar Tun Razak, Cheras, Kuala Lumpur 56000, Malaysia; chlorinelee91@gmail.com (Y.L.); abid@medcentral.com.my (A.N.)

**Keywords:** osteoarthritis, joint inflammation, hypoxic regulation mechanism

## Abstract

Osteoarthritis (OA) is one of the leading joint diseases induced by abnormalities or inflammation in the synovial membrane and articular cartilage, causing severe pain and disability. Along with the cartilage malfunction, imbalanced oxygen uptake occurs, changing chondrocytes into type I collagen- and type X collagen-producing dedifferentiated cells, contributing to OA progression. However, mounting evidence suggests treating OA by inducing a hypoxic environment in the articular cartilage, targeting the inhibition of several OA-related pathways to bring chondrocytes into a normal state. This review discusses the implications of OA-diseased articular cartilage on chondrocyte phenotypes and turnover and debates the hypoxic mechanism of action. Furthermore, this review highlights the new understanding of OA, provided by tissue engineering and a regenerative medicine experimental design, modeling the disease into diverse 2D and 3D structures and investigating hypoxia and hypoxia-inducing biomolecules and potential cell therapies. This review also reports the mechanism of hypoxic regulation and highlights the importance of activating and stabilizing the hypoxia-inducible factor and related molecules to protect chondrocytes from mitochondrial dysfunction and apoptosis occurring under the influence of OA.

## 1. Introduction

Osteoarthritis (OA) is a long-term chronic disease caused by the deterioration of cartilage in joints, resulting in bones rubbing together, which causes stiffness, pain and swelling that often impairs the movement of joints. It commonly affects the joints in the knees, although joints in other areas such as the shoulder, hip, hands and feet can also be affected [1]. Approximately 32.5 million of adults in the US are affected by OA [2]. The disease is more prevalent among the elder population, especially women. According to an analysis of 88 studies involving 10,081,952 participants, the global prevalence of OA among individuals aged 15 and above was 16.0%, while the prevalence of OA among individuals aged 40 and above was 22.9%. The ratio of OA incidence was higher among females (1.69) then males (1.39). The study also showed that the incidence of OA increased with age, which peaked at ages between 70 and 79 years old [3]. Besides age and gender, other risk factors of OA include joint injury and obesity. Individuals who have had joint injuries are more prone to developing OA. Meanwhile, the extra bodyweight in obese individuals may exert more stress on the joints, which tends to overuse the joints and results in OA [2]. Current treatment options for OA are limited. Apart from physiotherapy, regular exercise and weight management, pharmacological interventions are restricted to symptomatic relief with local intra-articular injections of corticosteroids and systemic administration of analgesics and non-steroidal anti-inflammatory drugs. In severe cases of OA, osteotomy, a surgical intervention performed to change the load pattern in the joint by altering leg alignment, may seem to be the only option to partially restore joint functionality. This option, however, is invasive and cannot fully restore joint function [4]. Hence, other treatment options such as restoring damaged chondrocytes are being investigated to treat OA.

OA is caused by an imbalanced redox state in the chondrocytes due to excessive production of nitric oxide (NO) and reactive oxygen species (ROS). The overproduction of NO and ROS due to damaged chondrocytes leads to mitochondrial dysfunction, which in turn results in poor differentiation and apoptosis of chondrocytes [5,6]. Diverse approaches are currently under research to treat OA by targeting NO and ROS, using biochemical products, such as nitric oxide synthase (NOS) inhibitors and superoxide dismutase (SOD). NOS inhibitors act to inhibit the production of NO in chondrocytes. For instance, the severity of OA in canine articular cartilage was decreased when treated with NOS inhibitor [7]. SOD is an enzymatic antioxidant that acts as an ROS scavenger by converting the superoxide anion radical (O_2_^−^) into hydrogen peroxide (H_2_O_2_) and oxygen, maintaining a balanced redox state in chondrocytes [8]. Thus, the levels of SOD were found to be significantly decreased in OA articular cartilage, indicating excessive ROS production [9]. Additionally, the use of SOD was reported to significantly reduce ROS production in OA articular cartilage [10].

Recent research has focused on the treatment of OA by targeting NO and ROS via a hypoxic approach. This approach is based on the rationale that the level of inflammation is related to the oxygen tension at the articular cartilage. Culturing ex vivo cartilage tissue under hypoxic conditions was shown to reduce the production of NO and ROS [11]. This hypoxic approach to OA treatment has become more popular, especially after the discovery of hypoxia-inducible factor (HIF), a central regulator of adaptive response to hypoxia in chondrocytes that is suggested to have chondroprotective properties and can maintain cartilage homeostasis [12]. Since OA is caused by inflammation due to the imbalanced redox state of chondrocytes resulting from excessive ROS and NO, the disease may be treated by reducing these components in the cells. One of the methods is by introducing a hypoxic environment to the chondrocytes or using bioproducts that can induce cell hypoxia in the articular cartilage. In this review, we report different approaches used to understand the mechanism of the hypoxic environment as a potential treatment for OA diseases using in vitro, ex vivo and in vivo models.

## 2. Pathophysiology of Osteoarthritis

OA is associated with inflammation of the joints. During the inflammation process, the accumulation of inflammatory mediators such as pro-inflammatory cytokines occurs in the OA articular cartilage [13]. Tumor necrosis factor α (TNF-α), interleukin 1β (IL-1β) and IL-6 are examples of pro-inflammatory cytokines that promote inflammatory responses in the synovial joints [14]. NO, which is another important inflammatory mediator promoting inflammation of the joints, originated from the synthesis of amino acid L-arginine [5]. Increases in the production of pro-inflammatory cytokines such as TNF-α and IL-1 also lead to increases in NO [15]. Accumulation of these inflammatory mediators results in mitochondrial dysfunction [16,17]. At the intracellular level, mitochondrion is an organelle that is involved in important cellular processes, including cellular differentiation and the regulation of apoptosis and autophagy. Thus, it plays an important role in maintaining chondrocyte homeostasis [18]. However, due to the accumulation of inflammatory mediators in the damaged OA articular cartilage, the mitochondrial function of the chondrocytes is severely damaged. As a result, homeostatic disturbances characterized by prolonged inflammation, increased apoptosis, increased catabolic activity, decreased autophagy, and decreased mitochondrial biogenesis, occur in chondrocytes [19]. Alterations in the mitochondrial structure result in reduced mitochondrial respiration and overproduction of ROS, leading to oxidative damage in chondrocytes [20]. ROS consists of oxygen free radicals, including O_2_^−^ and hydroxyl radical (OH^−^). The overproduction of ROS results in the increased production of pro-inflammatory cytokines and inflammatory responses [6]. ROS also induces chondrocyte apoptosis by causing damage to the mitochondrial DNA (mtDNA) and reducing the repair capacity of mtDNA [21]. The damage and mutation of mtDNA lead to the production of functionally impaired respiratory chain subunits in mitochondria. This prevents effective differentiation of chondrocytes and induces further ROS production, which then lead to cell apoptosis [22]. Damaged mtDNA, reduced mtDNA repair capacity and cell viability and increased apoptosis were detected in chondrocytes isolated from OA cartilage [23]. In addition to ROS, pro-inflammatory cytokines such as TNF-α and IL-1β are also found to induce mtDNA damage [24], leading to decreased mitochondrial energy production and DNA transcription [25]. In addition to ROS, NO can also disrupt mitochondrial function by inhibiting cytochrome oxidase in the mitochondria, leading to reduced production of the electron transport chain [26]. This increases the formation of O_2_^−^, further contributing to the production of ROS in chondrocytes [27].

Joint inflammation is also associated with chondrosenescence, an age-dependent deterioration of chondrocyte function [28]. During chondrosenescence, aging chondrocytes exhibit a highly pro-catabolic nature and prolonged inflammation occurs [29]. At the same time, the mitochondrial respiratory complex malfunctions, which increases ROS production, causing damage to mtDNA and compromising the mitochondrial protein function. As the mitochondrial function deteriorates, increased chondrocyte apoptosis and decreased autophagy occur. Hence, damaged chondrocytes in the OA joints cannot be replaced or repopulated [30]. Therefore, finding new approaches to reduce mitochondrial damage in chondrocytes would lead to new therapeutics targeting OA diseases.

## 3. The Role of Hypoxia in Defeating Osteoarthritis

Hypoxia plays a significant role in the metabolism of articular cartilage. It induces changes in proteoglycan synthesis, expression of extracellular matrix (ECM) components, growth factors, glucose transporters and ATP levels in chondrocytes [27].

Since oxygen molecules are involved in the formation of inflammatory mediator NO, the level of inflammation is associated with oxygen tension in the articular cartilage [26]. For example, the production of NO in the articular cartilage was significantly changed under hypoxic conditions when being exposed to pro-inflammatory cytokines. Incubation of cartilage explants for 72 h at low oxygen levels (1% O_2_) resulted in significantly reduced production of NO when being exposed to 10 ng/mL IL-1α or 10 ng/mL TNF-α as compared to samples incubated in 20% oxygen. On the contrary, reoxygenation with 20% oxygen in previously hypoxic cartilage explants exposed to IL-1α or TNF-α resulted in significantly increased production of NO [31]. Another study showed that NO production was significantly reduced at 1% oxygen compared to 20% oxygen when subjected to mechanical compression. The study indicated that hypoxic conditions can inhibit endogenous production of NO from the articular cartilage [11]. Following the reduction of NO, the formation of O_2_^−^ can be inhibited, reducing the amount of ROS in chondrocytes (Figure 1). With the reduction of NO and ROS, balanced redox states of chondrocytes can be maintained and joint inflammation can be reduced, thereby reducing chondrocyte apoptosis.

Another rationale of applying hypoxic treatment to OA is inspired from the native microenvironment of articular cartilage. Articular cartilage is avascular, alymphatic and aneural and characterized by low cellularity. Due to the absence of vasculature components, articular cartilage develops in hypoxic environments [32]. Under normal conditions, the oxygen concentrations in articular cartilage vary between 0.5 and 10%, with 6% oxygen in the superficial zone and less than 1% oxygen in the deep zone of the cartilage layer (Figure 2) [33]. Chondrocytes are the only cell type present in the articular cartilage. As the resident cells, chondrocytes develop specific mechanisms to maintain normal tissue function under hypoxic environments. One of the mechanisms is by inducing increased expression of cartilage ECM components [34]. It was found that culturing articular cartilage at a low oxygen concentration of 5% significantly increased proteoglycan and collagen synthesis compared to culture at 20% oxygen [35]. Moreover, the synthesis of hyaluronan was also significantly increased after culturing articular cartilage for 12 h at 5% oxygen compared to 20% oxygen [36]. Another mechanism involves the regulation of metabolic homeostasis by mitochondria in the chondrocytes. In healthy articular cartilage, mitochondria in the chondrocytes function normally under hypoxic environments by regulating metabolic homeostasis in the cells, generating ATP and having effective control over chondrogenesis. Thus, normal cell differentiation can be maintained and damaged chondrocytes can be consistently replaced through the process [37]. Hence, hypoxia may serve as another treatment option for OA based on the effects it has on restoring the chondrocytes.

## 4. Current Experimental Models Addressing Osteoarthritis Treatment by Hypoxia

Inducing features of OA in experimental models is important to obtain a better understanding of the disease, as well as to assess the responses to potential therapies. Experimental models can be induced with OA through surgical and chemical interventions and experimentally accelerated aging. Each of these methods of inducing OA has its own advantages and disadvantages. For instance, spontaneously occurring OA in models induced through experimental aging enables the following of OA development from early to late stages. However, the method is costly, time-consuming and involves more variability in the disease phenotype. Surgically or chemically induced OA on the other hand is less time-consuming and the disease manifestations induced through this method are less variable. However, surgical and chemical models of OA reflect more on post-traumatic OA alterations rather than spontaneous changes occurring in human OA. Although none of these methods of inducing OA can fully reproduce the features and symptoms of human OA due to the complexity and heterogeneity of the disease, these methods can still be used to induce OA in models for OA-related studies [4]. Summaries of the available experimental models addressing OA is illustrated in Figure 3.

### 4.1. In Vitro Models

The molecular mechanisms and pathways involved in joint physiology following hypoxic treatment of OA are commonly elucidated using in vitro models. One of these in vitro models involves a cell culture on a dish. Depending on the purpose of the study, cells can be cultured as monolayers or three-dimensional (3D) structures. Culturing cells as monolayers would facilitate the examination of phenotypic changes in the cells [38,39]. For example, chondrocytes are cultured as monolayers to examine changes in the chondrocyte phenotype under hypoxic conditions [40]. Another study also cultured human-adipose-derived mesenchymal stem cells (hATMSCs) as monolayers to assess the cells’ osteochondrogenic differentiation potential under hypoxic conditions [41]. However, the monolayer culturing system does not include proper cell–matrix interactions, meaning it fails to mimic the actual interactions occurring in human joints. This lack of dimensionality of the cell–matrix interactions disturbs chondrogenesis in the cartilage [42]. Culturing cells in a 3D structure would be advantageous, since the 3D environment allows an extended degree of cell–matrix interactions that favors the normal formation of the cartilage phenotype, resembling the features of real human cartilage. Therefore, as compared to monolayer cell culture, 3D culture systems are more preferred in in vitro models when used to study the effects of hypoxia on chondrocyte differentiation. For example, a study used a 3D micromass culture of primary human articular chondrocytes to study the interactions between the cells and glycosaminoglycan (ECM component) under hypoxia. The results from this study showed that hypoxia increased the glycosaminoglycan content in the primary human articular chondrocytes [43]. Another study used a 3D culture of hATMSCs to examine how the interactions between the cells and glycosaminoglycan, as well as type II collagen, affect the osteochondrogenic differentiation ability of the cells under hypoxia. Under hypoxia, the expression of chondrogenic markers, glycosaminoglycan and type II collagen were upregulated in hATMSCs [41]. Similarly, a separate study cultured chondrocytes on a 3D agarose culture to study the effects of hypoxia on the functional properties, including ECM production and chondrogenesis of cartilage tissue engineered using chondrocytes. Hypoxia was shown to promote ECM production and enhance chondrogenesis in the chondrocytes cultured in the 3D structure [44]. Therefore, using 3D models is recommended to better resemble the disease microenvironment and enhance the understanding of OA.

#### 4.1.1. Cells

Articular chondrocytes are commonly used cell types in in vitro models for studies related to the hypoxic treatment of OA. These cell types can be isolated from either human or animals. Human articular chondrocytes are preferred, as the phenotypic cell similarity provides a more accurate reflection of events that would likely occur in OA articular cartilage when exposed to hypoxic treatment. An example of a human articular chondrocyte is the human immortalized articular chondrocyte cell line C28/I2. This particular cell line expresses high levels of matrix-associated molecules that are involved in the catabolic and anabolic processes of articular cartilage. Therefore, it is a suitable in vitro model to be used for studies requiring the analysis of these matrix-associated molecules. A previous study used the cell line C28/I2 to analyze the expression of histone methyltransferase disruptor of telomeric silencing 1-like (DOT1L) and HIF molecules under hypoxic condition. An expression analysis of both DOT1L and HIF would enhance the understanding of the relationship between hypoxia and the molecules involved, as well as the mechanisms involved in the protection of chondrocytes against OA-related deterioration [43]. The convenience of maintaining human articular chondrocytes in hypoxic environments is another added advantage of the cell type to be used in vitro. A study showed that incubating the human articular chondrocytes for three days at 37 °C under hypoxic conditions of 1% oxygen tension supplemented with 5% carbon dioxide increased the anabolic and anticatabolic effects of the cells. This was shown through the upregulation of *SOX9*, a master regulator gene essential for cartilage development, and the downregulation of key cartilage-degrading enzymes, including aggrecanase 2/ADAMTS-5 and collagenase 3/matrix metalloproteinase 13 (MMP-13). The results showed a favorable effect of hypoxic treatment on human articular chondrocytes [45]. Consequently, the careful selection of cell types and their sources is critical and deemed important to understand OA-related pathologies.

While healthy human articular chondrocytes are difficult to obtain due to ethical issues, chondrocytes isolated from animal models such as murine and porcine models are used instead. In a study comparing hypoxic effects on the changes in chondrocytes isolated from different sources, porcine chondrocytes were shown to have similar responses to hypoxia as human chondrocytes, while murine chondrocytes were not shown to have similar hypoxic regulation effects as in human and porcine models. From the results, hypoxia induced anabolic effects on both human and porcine chondrocytes, as evidenced by the increases in *SOX9* expression, HIF and TIMP-3 molecules, a key anticatabolic factor, in the cells. These molecules were, however, not upregulated in murine chondrocytes, suggesting that mouse cells do not have a similar response to hypoxia as human and porcine cells. The absence of hypoxic regulation in murine chondrocytes may be due to the very thin lining of the mouse cartilage, meaning it is unable to produce a sufficient oxygen diffusion barrier, which is essential for the creation of a hypoxic microenvironment in the cells [45]. Despite the lack of hypoxic regulation in murine chondrocytes, they remain one of the popular cell types used in vitro to study the effects of hypoxia on OA. In fact, several studies have shown a different molecular response in murine chondrocytes when exposed to hypoxic environments. For example, in a study that used murine chondrocytes as an experimental model, it was shown that under hypoxic conditions of 3%, 2%, 1% and 0.5% oxygen, HIF (markers for hypoxic effects), SOX9 and type II collagen (markers for chondrocytes) were all significantly upregulated, while MMP-13 and ADAMTS-5 (markers for cartilage proteinases) were significantly downregulated. This suggested that hypoxia promoted a protective autophagy function in murine chondrocytes [40]. Similarly, another study also showed increased expression of HIF in murine chondrocytes under hypoxic conditions (1% O_2_). The stabilization of HIF activated the autophagy protective mechanism and prevented mitochondrial dysfunction in the cells, thereby preventing apoptosis and senescence of the murine chondrocytes [46]. However, the use of murine chondrocytes is not recommended, as they are not good representatives to study the disease, as mentioned later in this review.

Bovine articular chondrocytes show a similar positive response to hypoxia exposure as human articular chondrocytes. In a study where bovine articular chondrocytes were incubated with 2% oxygen in a 3D alginate culture for 17 days, it was observed that proteoglycan synthesis was increased after only 1 day of incubation. The increase was maintained for 17 days of hypoxic culture. Additionally, increases in the total amounts of type II collagen and glycosaminoglycan deposition were also observed in the bovine articular chondrocytes, which suggested that hypoxia enhances the matrix synthesis of the cells. Enhancement of the matrix synthesis would increase chondrocyte viability [47]. In addition to increasing the matrix synthesis, hypoxia also increases the tensile strength of engineered cartilage fabricated using bovine articular chondrocytes. A study discovered that treating bovine articular chondrocytes with 4% oxygen (hypoxia) on the 3rd and 4th weeks of culture before using them to construct neocartilage leads to a 2-fold increase in the tensile strength of the neocartilage. The greatly improved tensile strength was found to be associated with an increase in collagen–pyridinoline crosslinks, shown through the 20-fold increase in the expression of the lysyl oxidase (LOX) gene encoding enzyme responsible for the formation of collagen–pyridinoline crosslinks [48]. Hence, hypoxia not only enhances matrix synthesis, it also improve the tensile strength of the matrix formed in the chondrocytes. The increased tensile strength would help protect the chondrocytes from destruction, reducing the incidence of OA.

Mesenchymal stem cells (MSCs) are another choice of cell type to be used as in vitro models in studies related to the hypoxic treatment of OA. This cell type is rather popular and promising, especially for tissue engineering research related to OA treatment, due to the cells’ ability to differentiate and develop into different multilineages, including chondrocyte. MSCs can be isolated from humans as well as from animals [49]. A study used human-synovium-derived mesenchymal stem cells (hSDMSCs) to assess the effects of hypoxia on the chondrogenesis of the cells. The study results showed that exposure of hSDMSCs to 5% oxygen (hypoxic) significantly increased the proliferation and colony-forming characteristics of hSDMSCs as compared to 21% oxygen exposure (normoxic). Hypoxic culture of hSDMSCs also enhanced chondrogenesis within the course of 21 days in the presence of transforming growth factor β (TGF-β), which was indicated through the elevated chondrogenesis-related gene expression levels, including SOX9, type II collagen and aggrecan [50]. In another similar study of hypoxic effects on chondrogenesis, human-adipose-derived mesenchymal stem cells (hADMSCs) were used instead. It was shown that the level of hADMSC differentiation into the chondrogenic lineage was significantly increased under low 5% oxygen tension compared to normal 20% oxygen tension, as shown through the increase in type II collagen [41]. Furthermore, for human infrapatellar fat pad-derived MSCs (hIFPMSCs), the chondrogenic marker genes and transcription factors, including SOX5, SOX6, SOX9, type II collagen, type IX collagen, type XI collagen, aggrecan and versican, were all significantly enhanced under hypoxic condition (5% O_2_) compared to under normoxic conditions [51]. In addition, MSCs derived from human umbilical cord blood (hUCBMSCs) and bone marrow (hBMMSCs) cells were also used, wherein low oxygen tension was shown to have enhanced the expansion potential of hUCBMSCs [52] and the chondrogenic differentiation of hBMMSCs [53]. The therapeutic potential of human MSCs puts them ahead as strong candidates to treat OA by renewing the chondrocyte population and recruiting different cell types to tackle inflammation and restore normal cartilage function.

Apart from human MSCs, animal-derived MSCs are also commonly used in in vitro models in regenerative medicine research addressing OA. For instance, a study used rabbit bone marrow-derived mesenchymal stem cells (rBMMSCs) to investigate the effects of MSCs in combination with hyaluronic acid for OA treatment. The experiment showed that culturing rBMMSCs under 1% oxygen in chondrogenic, adipogenic and osteogenic media retained the rBMMSCs’ multipotency. The rBMMSCs were able to differentiate into chondrocytes, fat cells and osteoblasts, even under hypoxic conditions [54]. Another study used porcine-IFPMSC-encapsulated agarose hydrogel as a cartilaginous construct to examine the effects of low oxygen tension on the functional properties of the construct. Low 5% oxygen tension was observed to significantly promote ECM production in porcine IFPMSCs as compared to cells cultured under normoxic 20% oxygen. When compared to porcine chondrocytes cultured under the same hypoxic conditions, porcine IFPMSCs showed superior chondrogenic potential, leading to superior mechanical functionality in the resulting cartilage construct [44]. Overall, hypoxia exerts similar beneficial effect on MSCs of all origins and can induce the differentiation of MSCs into chondrocytes.

Lastly, osteoblasts are also used to investigate the influence of hypoxia on the regeneration ability of the cells. The results from a hypoxia experiment using human osteoblasts as a model showed that under low oxygen tension (2% O_2_) and supplementation of vitamin D, leptin production increased. Increases in leptin production induced bone remodeling, leading to the formation of an abnormal osteoblast phenotype and contributing to the pathophysiology of OA [55]. Hence, using different cell types helps to better understand OA and treat the disease.

#### 4.1.2. Ex Vivo Tissue

Other examples of in vitro model used in experiments of hypoxic treatments of OA are ex vivo tissue samples such as articular cartilage samples. Articular cartilage has become a candidate model because it has a key structural and functional roles in joint disease and physiology. It can be isolated from organisms and tested in experimentally controlled conditions to detect specific pathological hallmarks via biochemical and immunohistochemical analysis. Articular cartilage models are very useful in elucidating the contributions of different cartilage-degrading enzymes such as MMP-3 and MMP-13 [4]. For example, a study used articular cartilage samples isolated from humans, mice and pigs to compare the effects of hypoxia on cartilage protection through the analysis of cartilage-degrading enzymes, including ADAMTS-5 and MMP-13. Under hypoxic conditions with 1% oxygen, spontaneous cartilage degradation was significantly reduced in both human and porcine cartilage samples following downregulation of ADAMTS-5 and MMP-13 expression. Interestingly, hypoxia did not appear to reduce cartilage degradation in mouse cartilage, as the expression levels of ADAMTS-5 and MMP-13 were not upregulated. This may have been due to the absence of hypoxic regulation in mice because of the very thin lining in the mouse cartilage, which is insufficient to create a hypoxic microenvironment in the mouse chondrocytes. Hence, mice do not develop evolutionary adaption traits in response to chronic hypoxic stimuli to prevent destructive effects on chondrocytes, as shown in larger animals with thicker cartilage lining [45].

More functional studies can be performed using ex vivo tissue culture. For instance, slices of human OA cartilage or mice femoral head caps can be kept in culture to enable the analysis of the protein secretion profile in the presence or absence of specific factors that can mimic the OA-associated environment. These can also be useful in elucidating the contributions of different tissue-destructive enzymes in cartilage degradation, such as MMP-3 and MMP-13 [56]. Additionally, ex vivo tissue explants can also be used to replace in vivo models and to minimize the use of animals because they can mimic the behavior of the cells in ECM-like environments in biomechanical loading experiments, thereby assessing the effects of mechanical strain on chondrocytes [4].

### 4.2. In Vivo Models

For experiments involving a hypoxic intervention in OA treatment, in vivo models are mainly used to recapitulate the healing processes of OA-associated lesions in animals. Small- and medium-sized animals such as mice and rabbits are much preferred choices because they are cheap and easy to handle. For example, OA lesions can be easily induced on the animals via surgical approaches, including destabilization of the medial meniscus (DMM) in mice [46] and anterior cruciate ligament transection (ACLT) in rabbits [54].

A previous study investigated the effects of dimethyloxalyglycine (DMOG) and hypoxia on knee joint healing in a DMM-induced OA mouse model by observing the imageology and histomorphology of the joints. The experiment results showed erosion, hypo-cellularity of superficial articular cartilage and proteoglycan loss in knee joints of DMM mice. The DMM mice treated with DMOG under hypoxic stimulation (1% O_2_) showed a more complete cartilage surface and richer proteoglycan. This indicated the stabilization of HIF by DMOG and the hypoxic induction of a protective autophagy effect on the chondrocytes, which promoted cartilage formation in the mouse joints [46]. Another study used an ACLT-induced OA rabbit model to investigate the effectiveness of the intra-articular injection of MSCs cultured under hypoxic conditions (1% O_2_) in combination with hyaluronic acid (HA) as a therapeutic option for OA. Marked gross changes in OA, including cartilage abrasion, osteophyte formation and subchondral bone exposure, were observed on the knee joints of ACLT rabbits without MSC + HA treatment. On the contrary, ACLT rabbits that received MSC + HA treatment under hypoxic conditions showed diminished gross changes of OA on the joint surfaces [54]. As such, the combined effects of hypoxia and ECM proteins are important to help regenerate the cartilage through hypoxic regulation of the microenvironment and by regulating cell oxygen uptake.

Evidently, larger animals such as dogs [57], pigs [58] and horses [59] are more suitable in vivo models, especially for studies involving age-related human OA onset. This is because these animals can develop spontaneous OA without surgical or chemical interventions, which is more translatable for age-related OA in humans. However, ethical issues and the long timescales for OA development with these animals prevent their usage in experimental settings. All of these issues would make studies prohibitively expensive and ultimately unfeasible. As such, they are not featured in any hypoxia-related OA studies. In general, every experimental model has their respective advantages and disadvantages. Thus, it is important to choose a suitable model by taking into consideration the study objectives.

## 5. Mechanisms of Hypoxic Regulation

By reviewing all of the experimental models stated above, it was noted that chondrocytes in the joint cartilage develop an adaptive response that can maintain proper cell function and prevent apoptosis in hypoxic environments. Hypoxia-inducible factor (HIF) is the central regulator of this adaptive response to hypoxia through the transcriptional activation of genes modulating oxygen homeostasis (Figure 4). The heterodimer of HIFs includes HIF-α (HIF-1α, HIF-2α and HIF-3α) and HIF-β subunits [60]. Under normal oxygen conditions, HIF-α binds to the von Hippel–Lindau (VHL) protein and the ubiquitin ligase system is activated, leading to the proteasomal degradation of HIF-α [61]. Prolyl hydroxylase (PHD) is an important player in the hydroxylation of proline residues in HIF-α, which are essential for VHL binding [60]. Under hypoxic conditions (oxygen tension < 5%), the activity of PHD is suppressed, resulting in the stabilization of HIF-α and its dimerization with HIF-1β. The stabilized HIF-α then translocates to the nucleus and binds to hypoxia response element (HRE) within the promoter region, and genes that control cellular oxygen homeostasis are activated [62]. Hypoxic environment in the articular cartilage stabilizes HIF-1α activity in the chondrocytes [26]. This is proven by the increase in HIF-1α expression in chondrocytes cultured under 1% oxygen [40,46]. HIF-1α exerts chondroprotective effects on the cells and maintains cartilage homeostasis [12]. It has been shown to alleviate OA by maintaining ECM integrity in articular cartilage through enhanced mitophagy. Through the increase in BNIP3 expression in a dose-dependent manner following elevated expression of HIF-1α, it has been indicated that the HIF-1α/BNIP3 signaling pathway is involved in the protective role of mitophagy [46]. Furthermore, HIF-1α can mediate anticatabolic responses and inhibit articular cartilage degradation [45]. A study showed that the anticatabolic function of HIF-1α in maintaining intact articular cartilage occurs through the suppression of the NF-κB signaling pathway [63].

Besides acting alone, HIF-1α also cooperates with other factors such as microRNA (miRNA) to protect the articular cartilage from destruction in OA. The miRNA is a small non-coding RNA molecule that functions in RNA silencing and the post-transcriptional regulation of gene expression [64]. Like HIF-1α, miRNA plays an important regulatory role in hypoxia regulation [65]. An example of a hypoxia-regulatory miRNA is miRNA-146a. A previous study found that HIF-1α induced miRNA-146a under hypoxia and miRNA-146a promoted autophagy in chondrocytes by decreasing Bcl-2 (autophagy inhibitor) expression [40]. In addition to miRNA, HIF-1α also induces the histone methyltransferase disruptor of telomeric silencing 1-like (DOT1L), which encodes an enzyme that methylates lysine 79 of histone H3 (H3K79) and is involved in epigenetic regulation of transcription [66]. Under hypoxic conditions, DOT1L was induced by HIF-1α and exerted protective effects on articular chondrocytes via the H3K79 methylation pathway. Intra-articular treatment with IOX2 (hypoxia exposure) halted OA progression in mice, as DOT1L and H3K79 methylation were restored in the articular cartilage [43].

Unlike HIF-1α, which has a definite chondroprotective effect on articular cartilage, another heterodimer of HIF, HIF-2α, plays a controversial role in the homeostasis of chondrocytes. Several studies have highlighted this role of HIF-2α in OA pathogenesis [67,68]. It was found that HIF-2α had an abnormal expression pattern in OA cartilage and that it regulated the hypertrophic differentiation of OA chondrocytes. This was shown through the overexpression of the Epas1 gene associated with HIF-2α observed in the degradation of knee cartilage [67]. Another study reported that HIF-2α causes OA by directly promoting the expression of cartilage-degeneration-related genes, including MMP-3 and MMP-13 [68]. In addition to acting alone, HIF-2α has been shown to interact synergistically with vitamin D to produce leptin, an element that induces OA in articular cartilage types [55]. Additionally, a recent discovery that miRNA-455s can suppress the expression of HIF-2α and reverse the OA conditions further supported the role of HIF-2α in OA pathogenesis [69]. However, more genetic studies involving larger cohorts of multiple ethnic populations suggested that HIF-2α is not associated with OA pathogenesis [70,71]. Another different study even reported decreased levels of HIF-2α expression in OA cartilage [72]. Moreover, the inconsistency of HIF-2α expression in response to inflammatory cytokines induced by inflammation in the damaged cartilage further complicates its role in OA pathogenesis. For example, one study reported elevated levels of HIF-2α induced by the inflammatory cytokine IL-1 in mouse articular chondrocytes [68]. On the contrary, another separate study showed decreased levels of HIF-2α in human articular chondrocytes under IL-1 stimulation [45]. Even though the difference in HIF-2α expression between the two studies may be attributed to the fundamental differences in HIF-2α responses between human and murine models, it is undeniable that the role of HIF-2α in OA pathogenesis is still confusing and that more research is required to confirm its association with chondrocyte homeostasis.

Since the role of HIF-2α in the maintenance of chondrocytes remains ambiguous while the chondroprotective role of HIF-1α is clear, HIF-1α has become the more obvious target of intervention in OA treatment. Evidence from the studies mentioned above emphasized the stabilization of HIF-1α as a key point to protect the articular cartilage from OA destruction. Thus, in addition to hypoxic intervention, a PHD inhibitor such as DMOG or its analog 2-oxoglutarate may be used together to alleviate OA. However, a study that used both DMOG and 2-oxoglutarate in cultures of hypoxic human chondrocytes revealed an adverse effect of treatment with the 2-oxoglutarate, whereby the secretion of type II collagen was halted [73]. This suggests the specificity of DMOG in regulating osteoarthritis. DMOG was found to have increased HIF-1α expression, activated autophagy, inhibited mitochondrial dysfunction and protected chondrocytes against apoptosis. The treatment of OA mice with DMOG was also found to have ameliorated OA development [46]. Therefore, the PHD inhibitor can be used as a supplement to hypoxic treatment to manage OA in a more effective way.

## 6. Concluding Remarks and Future Prospective

OA is associated with inflammation of the joints caused by the accumulation of pro-inflammatory mediators and an imbalanced redox state in chondrocytes due to excessive production of NO and ROS. This imbalance can be corrected through hypoxic treatment. By reviewing different experimental models used in studies related to the hypoxic treatment of OA, the mechanisms of hypoxic regulation in articular cartilage can be understood. With this understanding of hypoxic regulation mechanisms, effective therapeutic strategies may be devised to treat OA. In this review, the stabilization of HIF-1α appears to be the key point in hypoxic regulation. Therefore, PHD inhibitors such as DMOG may be used in combination with hypoxic interventions to treat OA. Furthermore, with an understanding on the characteristics of different types of experimental models, a suitable model can be selected depending on the purpose of the research.

In the future, a focus should be placed on researching the stabilization of HIF-1α through the use of PHD inhibitor. Research on the stabilization of HIF-1α should be carried out in different models to find out how this condition can be translated into actual clinical trials in OA patients. Moreover, the safety and efficacy of the PHD inhibitor should also be examined before being used for OA treatment.

## Figures and Tables

**Figure 1 ijms-23-05356-f001:**
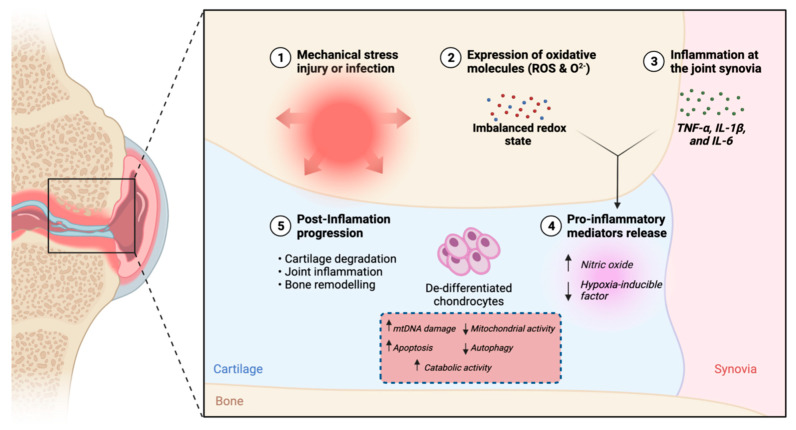
The consequences of inflammation of the articular cartilage inducing the production of NO, O_2_^−^ and ROS under normoxia conditions, leading to severe joint dysfunction and blocking movement. Adapted from “Pathology of Osteoarthritis” by BioRender.com (2022). Retrieved from https://app.biorender.com/biorender-templates (accessed on 4 April 2022).

**Figure 2 ijms-23-05356-f002:**
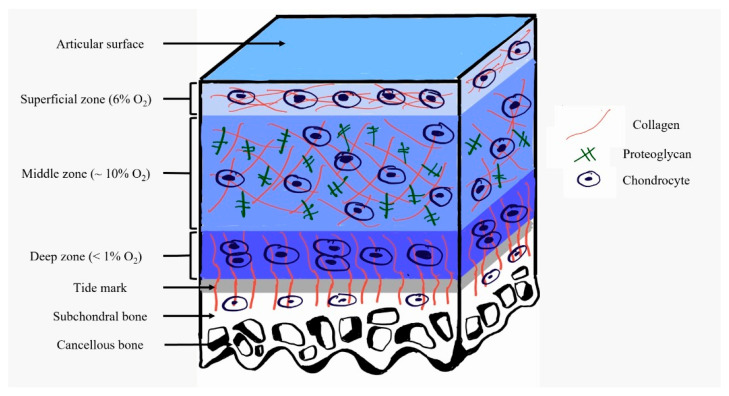
Variations in oxygen concentration in different regions of the articular cartilage.

**Figure 3 ijms-23-05356-f003:**
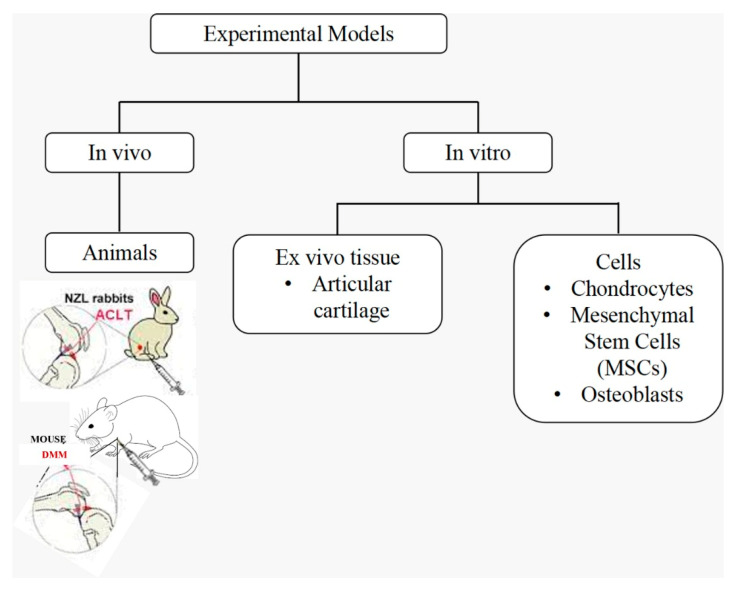
Summary of experimental models addressing OA treatment using hypoxia.

**Figure 4 ijms-23-05356-f004:**
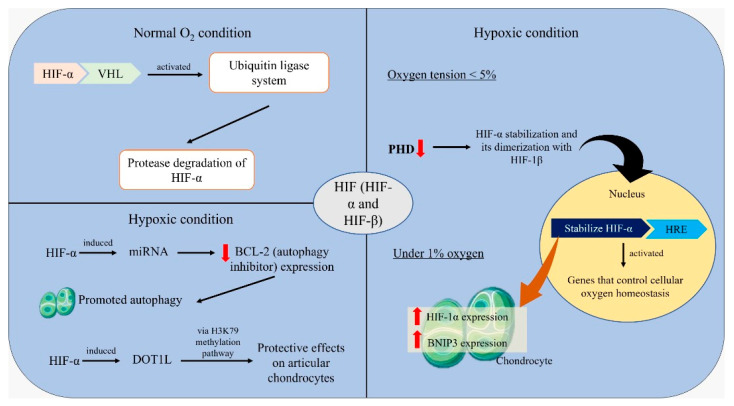
Illustration of mechanisms of hypoxic regulation.

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
