# Peer review of "Remodeling Osteoarthritic Articular Cartilage under Hypoxic Conditions"

_ijms, 2022, doi:10.3390/ijms23105356_

Round 1

Reviewer 1 Report

Thanks for the opportunity to review the manuscript entitled “Modeling articular cartilage osteoarthritis under hypoxic conditions”. The authors have done an impressive work to summarize the latest understanding of hypoxia in osteoarthritis. Some minor issues need to be addressed before publication.

  1. What’s the physiologic function of hypoxia in knee articular cartilage?

  1. What’s expression pattern of hypoxic gene, such as HIF pathway in the osteoarthritic articular cartilage?

  1. The title should be remodeling osteoarthritic articular cartilage under hypoxic conditions

  1. Page 9, “Under the hypoxic conditions-----, spontaneous cartilage degradation was significantly reduced in both human and porcine cartilage, following upregulation of ADAMTs-5 and MMP-13 expression”. Don’t you think the cartilage degradation and ADAMTs and MMPs expression was inconsistent? What’s the explanation? And same goes for the following sentence ”interestingly -------”.

Reviewer 2 Report

Dear Author,

This is an outstanding review report, concise and well written.

Best regards

Reviewer 3 Report

The manuscript entitled "Modeling Articular Cartilage Osteoarthritis Under Hypoxic Conditions" is clear, well structured and relevant to the field. The authors describe the role of NO and ROS in the pathophysiology of OA and thus highlight hypoxia regulation as a potential efective strategy for OA treatment. A major strength of the manuscript is that it provides a comprehensive review on the in vitro models used to investigate the role of hypoxia for OA treatment. A main point for optimisation is the discussion on the role of prolyl hydroxylase domain enzymes for regulation of HIF and the effect of PHD inhibitors in different OA models/model systems. Since one of the concluding remarks in the manuscript defines combination of hypoxic intervention and PHD inhibitor as DMOG as potential treatment for OA, the review of research on DMOG (and other PHD inhibitors) effectiveness should be strenghtened. In addition, the following points should be addressed before publication:

  1. Revise figure 2. This figure is very similar to an image found online (Articular Cartilage - Basic Science - Orthobullets; https://teambone.com/education/education-clinical/orthopedic-basic-science/articular-cartilage-anatomy/). Moreover, my major concern is that this figure does not appropriately illustrate the cellular organization in the zones of articular cartilage. It needs to be improved.
  2. Correct the cursive font-style for the terms "in vivo" and "in vitro". It should be italic.
  3. Subsection 4.1.2., fifth sentence: replace "murine and porcine" with "mouse and pig".
  4. Figure 3 - include representative examples for animal models for OA.
  5. Include a figure that illustrates the mechanisms of hypoxic regulation discussed in section 5.

Reviewer 4 Report

This article is a valuable review of whether the hypoxic response of articular cartilage is useful in the treatment of osteoarthritis.

The description of HIF focuses on the chondroprotective effects of HIF-1alpha. However, the role of HIF-2alpha in the osteoarthritis pathogenesis should also be adequately described. From the following publications in 2010, various studies have been conducted targeting HIF-2alpha and should be reviewed

Saito T. Transcriptional regulation of endochondral ossification by HIF-2alpha during skeletal growth and osteoarthritis development. Nat Med. 2010 Jun;16(6):678-86.

Yang S. Hypoxia-inducible factor-2alpha is a catabolic regulator of osteoarthritic cartilage destruction. Nat Med. 2010 Jun;16(6):687-93.

Reviewer 5 Report

It is a good review of literature, well structured and I think useful for the researchers in this field. 

Round 2

Reviewer 4 Report

This revised manuscript addresses the reviewer's concerns. It deserves attention and is worthy of publication in this journal.